# Development of a Wound-Healing Protocol for In Vitro Evaluation of Urothelial Cell Growth

**DOI:** 10.3390/mps6040064

**Published:** 2023-07-05

**Authors:** Christopher Foster, Todd Jensen, Christine Finck, Courtney K. Rowe

**Affiliations:** 1Department of Pediatrics, University of Connecticut School of Medicine, Farmington, CT 06032, USA; 2Division of Pediatric General and Thoracic Surgery, Connecticut Children’s, Hartford, CT 06108, USA; 3Division of Pediatric Urology, Connecticut Children’s, Hartford, CT 06108, USA

**Keywords:** urethra, wound healing, growth factor, regenerative medicine

## Abstract

Urethral healing is plagued by strictures, impacting quality of life and medical costs. Various growth factors (GFs) have shown promise as therapeutic approaches to improve healing, but there is no protocol for in vitro comparison between GFs. This study focuses the development of a biomimetic in vitro urothelial healing assay designed to mimic early in vivo healing, followed by an evaluation of urothelial cell growth in response to GFs. Methods: Wound-healing assays were developed with human urothelial cells and used to compared six GFs (EGF, FGF-2, IGF-1, PDGF, TGF-β1, and VEGF) at three concentrations (1 ng/mL, 10 ng/mL, and 100 ng/mL) over a 48 h period. A commercial GF-containing medium (EGF, TGF-α, KGF, and Extract P) and a GF-free medium were used as controls. Results: There was a statistically significant increase in cell growth for IGF-1 at 10 and 100 ng/mL compared to both controls (*p* < 0.05). There was a statistically significant increase in cell growth for EGF at all concentrations compared to the GF-free medium control (*p* < 0.05). Conclusion: This study shows the development of a clinically relevant wound-healing assay to evaluate urothelial cell growth. It is the first to compare GFs for future use in reconstructive techniques to improve urethral healing.

## 1. Introduction

Male urethral strictures severely impact the quality of life of patients and their caregivers [1,2]. Strictures can occur following pediatric urologic surgery to treat congenital differences such as hypospadias, epispadias, or bladder exstrophy. They can also occur in adulthood following surgery for benign or malignant prostate disease. Finally, they can result from pelvic trauma from a blunt or penetrating injury or urethral trauma due to catheter placement. Urethral strictures account for 1.5 million office visits a year and an estimated 191 million USD in health care costs per year. Patents with strictures experience complications such as infections, bladder stones, fistulas, and renal failure [3]. While surgical repair in skilled hands can go well, a number of techniques and diagnoses have poor success rates. The success rate for incising a urethral stricture—internal urethrotomy—is 20–30%, while proximal hypospadias repairs have a complication rate of 32–68% following surgery [4,5]. There is a compelling need for continued research into urethral healing to prevent complications such as strictures after surgery or injury.

Regenerative medicine has provided enticing preliminary options for urethral replacement and repair. The key element in these approaches are growth factors (GFs), signaling molecules that can stimulate the growth of cells during wound healing [6]. GFs such as VEGF, IGF-1, EGF, and TGF-B have preliminary results showing their addition to urethral repair reduces strictures in animal and in vitro models (Table 1). Unfortunately, the study methods have been so varied that there is no consensus on the GF, concentration, location of delivery, or timing of delivery needed to impact urethral healing. In part, this is due to the lack of an in vitro model that could be used to compare GFs in a controlled setting.

The goal of this study was to develop a protocol for evaluating early in vitro urothelial cell growth that mimics the urethral healing seen in vivo. A wound-healing assay was selected, as this is a well-established technique for studying the migration and growth of cells in vitro [7]. A wound or scratch is created in a cellular layer. Cells from the edges of the wound will migrate and fill in the gap. Images are taken after the initial wound is created, and then at subsequent time points. These images are compared to determine the rate of healing. Healing is defined as the percent of wound gap covered by urothelial cells.

The wound-healing assay was chosen as the best in vitro strategy to mimic early in vivo urethral healing. Lopes et al. studied urethral healing in pigs, Hafez et al. in rabbits, and Hofer in rats [8,9,10]. All found that the urethra first heals after an injury or surgery via ingrowth from the edges of the defect with a single layer of urothelial cells [8,9,10]. This ingrowth occurs over the first 48 h until the wound is covered in a single layer of urothelium. This pattern of urothelial ingrowth is mimicked by the wound-healing assay described in our current study. This allows for a clinically relevant experimental model to evaluate the impact of various GFs on early urothelial cell growth. Our protocol will allow for further studies that can serve as a backbone for future regenerative medicine techniques.

**Table 1 mps-06-00064-t001:** Summary of growth factors present in the healing urethra.

Growth Factor	Evidence in Urethra and Wound Healing
Epidermal growth factor (EGF)	EGF receptor increases keratinocyte proliferation and cell migration leading to re-epithelialization in wound healing [11]. EGF is crucial for urethral and penile development, and is deficient in the skin adjacent to the urethra in boys with hypospadias, a congenital urethral defect [12,13,14]. The addition of EGF improves urothelial cell healing in an in vitro model of bladder injury repair [15].
Fibroblast growth factor-basic (FGF-basic)	FGF-basic (also known as FGF-2) also plays a role in wound healing by increasing granulation tissue formation, re-epithelialization, and tissue remodeling [11]. It is present in the developing urethra and required for proliferation of urethral progenitor cells of the epithelium [16].
Insulin-like growth factor-1 (IGF-1)	IGF-1 has been linked to wound healing by increasing keratinocyte motility and promotes a proliferative response in the wound [11]. The IGF-1 receptor is prominently found in the epithelium of the rat urethra [17]. IGF-1 has been shown to promote urothelial cell proliferation resulting in improved urethral wound healing via stricture prevention [18].
Platelet-derived growth factor (PDGF)	PDGF has a role in wound healing by increasing the expression of VEGF and IGF-1 to improve angiogenesis and re-epithelialization. PDGF also increases the proliferation and stimulation of fibroblasts [11]. Additionally, PDGF-BB has already been approved by the FDA for topical wound treatment in diabetic ulcers [19,20].
Transforming growth factor beta (TGF-β1)	TGF-β1 has been proven to promote acceleration of healing, and to have anti-scarring and anti-fibrotic effects [11,21]. The TGF- β1 receptor is found on the mouse genital tubercle during development, and receptor expression has decreased levels after urethral injury, so increased TGF-β1 may promote urethral healing [22,23].
Vascular endothelial growth factor (VEGF)	VEGF mediates angiogenesis by improving tissue ischemia and hypoxia, and limiting fibrosis and stricture [24]. VEGF receptor expression is decreased in urethral subepithelia; however, it has been shown that increased VEGF in urethral tissue supports urethral repair [22].

## 2. Materials and Methods

### 2.1. Cell Selection

The cells used for the protocol needed to be clinically relevant and reliably cultured in the lab. The inner layer of the urethra is made up of urothelium, a specific type of stratified transitional epithelium. Urothelium is found lining the renal pelvis, ureter, bladder, and kidney, and is one of the eight epithelial tissue types found in the body [25]. While the term “urothelium” is often used interchangeably, the urothelium of the renal pelvis and ureter is derived from mesoderm, while the urothelium lining the bladder and urethra is derived from ectoderm [26]. It is therefore not surprising that the urothelial cells of the bladder and urethra are quite similar, with the exception that urothelial cells derived from the bladder show higher growth potential in culture [25]. This higher growth has led to urothelial cells from a bladder source being used as the gold standard for urethral replacement and cell-seeded urethral scaffolds in regenerative medicine [27,28,29,30,31,32,33]. Our protocol follows this standard, using urothelial cells from a bladder source (bladder epithelial cells) for the wound assay.

### 2.2. Cell Expansion

Primary bladder epithelial cells (PCS-420-032, ATCC, Manassas, VA, USA) were purchased and expanded using the provided protocol. Briefly, cells were transferred to a T75 flask at 4000 cells per cm^2^ with Prostate Epithelial Basal Medium (ATCC, Manassas, VA, USA) supplemented with Corneal Epithelial Cell Growth Kit (ATCC, Manassas, VA, USA) and Penicillin–Streptomycin–Amphotericin B Solution (ATCC, Manassas, VA, USA) at 5 mL of complete growth media per 25 cm^2^. Every 48 h, spent medium was removed and fresh medium was added until approximately 80% confluence was reached. Upon reaching approximately 80% confluence, spent medium was removed and cell layer was detached using 0.05% Trypsin–EDTA (Gibco, Thermo Fisher Scientific, Waltham, MA, USA). Trypsin–EDTA was neutralized with equal volume fetal bovine serum (Corning, Corning, NY, USA). Cells were collected and centrifuged (Sorvall ST8, Thermo Fisher Scientific, Waltham, MA, USA) for 5 min at 150 g. Cells were counted and either frozen in working volumes with 10% dimethyl sulfoxide (Thermo Fisher Scientific, Waltham, MA, USA) or used for wound-healing assay at passage 4.

### 2.3. Selection of Evaluated GFs and Controls

Six candidate GFs were selected based on their role in urethral or wound healing (Table 1). VEGF, PDGF, and TGF-β are also elevated in the rat urethra during healing after surgery [10]. Each GF was added directly to cell media as described below. GF concentrations were chosen based on preliminary trials within our lab to allow demonstration of a wide range of dose-dependent responses.

Two controls were used. GF-containing medium describes the combination of commercially available Prostate Epithelial Basal Medium (ATCC, Manassas, VA, USA) and Corneal Epithelial Cell Growth Kit (ATCC, Manassas, VA, USA) that are sold by American Type Culture Collection to be used together for growth of their cells. The Corneal Epithelial Cell Growth Kit contains EGF, TGF-α, KGF, and Extract P; specific quantities are proprietary. This was considered the control that best mimicked physiologic conditions. We also used a GF-free medium to have a control that did not contain any GFs. This allowed for direct comparison of GFs.

### 2.4. Wound-Healing Assay

Cells were inserted into the wells of a wound-healing assay culture-insert (ibidi Inc., Fitchburg, WI, USA) at a concentration of 6 × 10^5^ per mL of medium (42,000 cells for 70 µL total of medium). Cells were allowed to reach near confluence, which took approximately 20 h. At a time of 4 h prior to the start of the assay, cells were rinsed and 70 µL of fresh CnT-Prime Basal, Growth Factor Free Media (Zenbio Inc., Research Triangle Park, NC, USA) was added into the wells. At time 0 of the assay, spent medium was aspirated and the well insert was removed from the dish. The dish was rinsed with PBS and 2 mL of fresh medium (Zenbio growth factor free medium or ATCC epithelial basal medium) was added to dish. GFs EGF, FGF-basic, IGF-1, PDGF-BB, TGF-β1, and VEGF (PeproTech, Rocky Hill, NJ, USA) were added to their respective wells at concentrations of 1 ng/mL, 10 ng/mL, or 100 ng/mL. These concentrations were determined through previous trials and chosen to best illustrate the changing cell growth in response to changing GF concentrations.

At time points 0, 6, 10, 14, 24, and 48 h, 3 pictures of each wound were taken using an Evos XL Core (Thermo Fisher Scientific, Waltham, MA, USA). Upon completion of assays, the percent area of the wound that was not covered by cells was determined using ImageJ with the scratch assay analyzer function from the MiToBo plugin (Figure 1) [34]. The plugin provided the total area not covered by cells. The total area at subsequent time points was divided by the average of the initial time point 0 to determine percent remaining. The difference between 100 and area remaining was defined as the percent healing. A total of 12 measurements were taken for each GF and concentration.

Graphical analysis was performed using GraphPad Prism 9. Statistical analysis was performed using Excel. A two-tailed t-test with 2 sample unequal variance was performed to compare mean area of cell growth of experimental groups to the GF-free medium control. The GF-free medium was chosen as the control to better evaluate the impact of individual GFs. A *p*-value < 0.05 was considered statistically significant.

### 2.5. Qualitative Morphology

All analyses were performed using ImageJ, and 10 ng/mL concentration images were used. Morphological changes to the cells were quantified using a predetermined grid to identify 24 cells. These cells were outlined and measured for area, perimeter, aspect ratio, and circularity. We calculated the cells’ form factor (FF) by using equation as follows [35]:FF = 4πArea/Perimeter^2^

Form factor closer to 1.0 indicates a circular object. Statistical analysis was performed using Excel. A two-tailed t-test with two sample unequal variance was used to compare mean FF for experimental groups compared to the GF-containing medium control. The GF-containing medium was chosen as the control to compare results to the most physiologic growth. A *p*-value < 0.05 was considered statistically significant.

### 2.6. Gene Expression

RNA was isolated from collected cells following wound-healing assays. Using iScript gDNA clear synthesis kit, cDNA (Biorad, Hercules, CA, USA) was created and used in Epithelial to Mesenchymal Transition PrimePCR pathway plates (Biorad, Hercules, CA, USA). These commercially available plates evaluate 96 genes associated with epithelial to mesenchymal transition (EMT). Fold change was calculated using GAPDH as a reference gene and the GF-containing medium as the control. The GF-containing medium was chosen as the control to compare results to the most physiologic growth. Graphical and statistical analysis was performed using GraphPad Prism 9. To rapidly compare the six experimental groups to the control, we used a one-way ANOVA. In addition, a follow-up test using a Dunnett’s multiple comparison to compare each mean to the control mean was performed. A *p*-value < 0.05 was considered statistically significant.

## 3. Results

### 3.1. Wound-Healing Assay

The GF-containing medium had a statistically significant increase (*p* < 0.00001) in percent healing compared to the GF-free medium at 48 h (Figure 2). IGF-1 at concentrations of 10 and 100 ng/mL had a statistically significant increase (*p* < 0.05) in percent healing compared to both of the controls with 100 ng/mL, resulting in 100% healing at 48 h (Figure 3A). EGF at all concentrations of 1, 10, and 100 ng/mL (*p* ≤ 0.01) had a statistically significant increase in percent healing compared to the GF-free medium (Figure 3B). FGF-basic, PDGF-BB, TGF-β1, and VEGF all showed increased percent healing up to 48 h, but there was no statistically significant difference when compared to the GF-free medium (Figure 4).

### 3.2. Morphology

EGF showed elongation of cells and a loss of cell-to-cell adhesion, concerning for EMT (Figure 5A). FGF-basic, IGF-1, PDGF-BB, and VEGF retained a more uniform epithelial-like morphology, appropriate for urothelial cells. TGF-β1 showed a more rounded appearance (Figure 5E). A summary of phenotypic changes is presented in Table 2. The form factor of EGF (*p* = 0.025) and TGF-β1 (*p* = 0.03) had a statistically significant change compared to the GF-containing medium control. EGF had a form factor of 0.612 indicating a more abnormal, elongated shape. TGF-β1 had a form factor of 0.835 indicating a much more rounded morphology.

### 3.3. Gene Expression

EGF, FGF-basic, IGF-1, PDGF-BB, and VEGF showed no significant changes in gene expression compared to the GF-containing medium control. TGF-β1 showed numerous significant changes. A total of 14 genes had a statistically significant change when TGF-β1 was compared to other GFs and the control (*p* < 0.05). Highlighted are the 10 most significant genes (Figure 6).

## 4. Discussion

This study compares the impact of common GFs on in vitro urothelial cell growth using a wound-healing assay designed to mimic early urethral healing. IGF-1 and EGF both have a statistically significant effect on proliferation and cell growth without evidence of mesenchymal transformation. FGF-basic, PDGF-BB, TGF-β1, and VEGF had no significant difference when compared to GF-free medium control at all concentrations. TGF-β1 also showed evidence of epithelial to mesenchymal transition. To our knowledge, this is the first study that directly compares numerous GFs and their impact on urothelial cell growth using a clinically applicable in vitro model. These results can be used to guide further techniques to improve urethral healing.

Prior researchers have evaluated the impact of GFs on urothelial healing, but GF selection and methods have been too varied to provide meaningful comparisons. Shinchi et al. found that a catheter wrapped in collagen that permitted slow release of IGF-1 improved urethral lumen diameter after an electrocautery urethral injury in male rabbits [18]. Zhu et al. found that a gelatin scaffold with slow-released EGF and mitomycin C increased growth of urothelial cells in vitro without promoting fibroblastic growth [36]. Guan et al. modified urinary stem cells to express human VEGF and found improved neovascularization when cells were used to seed a decellularized graft in nude mice [28]. Jia et al. used a collagen graft containing VEGF for urethral replacement. They found that while strictures developed in both the VEGF group as well as the control group using just collagen, they were less severe in the VEGF group [37]. Wang et al. took a slow-release approach by incorporating VEGF within polylactic acid–glycolic acid nanospheres. Results showed improvements in stenosis rates months after urethral injury repair with bladder acellular matrix graft when compared to empty nanospheres or the graft alone [24]. A follow-up study by this group demonstrated increased angiogenesis behind these improved outcomes [38]. Meanwhile, a case report described using platelet-rich fibrin—which generally contains a mix of TGF-B, VEGF, and EGF—placed directly on the urethra of a single pediatric patient who subsequently developed no fistula, an intriguing though far from conclusive report [39]. Overall, the impact of GFs on urethral healing demonstrates promise for future therapies, but prior to this study there has been no opportunity to compare the impact between GFs.

Our initial hypothesis was that EGF would show the greatest impact on urothelial cell growth in vitro. EGF has long been linked to human urethral and penile development [12,14]. Zhu et al. found that the addition of EGF appeared to improve urothelial growth in a bladder injury model in vitro as well as urethral repair in a rabbit model [15,36]. The limited impact of EGF in our study may be explained by the early phase of urethral healing that our protocol was designed to mimic. Vinter-Jensen et al. presented data that suggest that EGF increased the volume of urothelium and submucosa in the ureter and bladder of rats over a 28-day period [40]. It may be that EGF has less impact on cellular growth in the early phase of urethral healing, but more impact in later phases, especially the proliferation phase that occurs between day 6 and day 10. It was also interesting to note that there was no significant dose-dependent increase in % healing with EGF, with fairly similar response at 10 ng/mL and 100 ng/mL, which was only significantly increased when compared to controls at the 48 h time point. It is possible that there is a ceiling limit to the impact of EGF on early urethral cell proliferation in vitro.

Fibrosis is the result of an excessive accumulation of extracellular matrix components caused by fibroblast cell response to cellular damage. These fibroblasts are derived from mesenchymal cells and epithelial cells through an EMT. EGF has been shown to induce this EMT. Cells transition from a homogenous cobblestone-like appearance to an irregular appearance with a loss of cell–cell adhesion [41]. Our study showed significant phenotypic changes in EGF representing a more abnormal and elongated shape compared to the normal urothelial phenotype, concerning for EMT. However, the gene expression analysis does not indicate this transition. Due to our study only encompassing 48 h, it is possible that these cells were simply proliferating and filling in the wound gap before regaining their tight junctions and cobblestone-like appearance. Despite the increase in cell growth, this risk of mesenchymal transition would likely make EGF a poor candidate for clinical interventions designed to support healthy urethral healing in vivo.

The impact of IGF-1 on early urethral cell growth was greater than anticipated, and unlike with EGF, the cells exposed to high levels of IGF-1 showed no morphological changes or evidence of mesenchymal transition. The strong growth seen with IGF-1 may be due to its known effect on epithelial cell proliferation in vitro, which Shinchi et al. showed translated into in vivo improvement in urethral healing by promoting urothelial cell proliferation. That study found that slow-release IGF-1 reduced formation of urethral strictures in rabbits [18]. Interestingly, IGF-1 has also been evaluated for use in other areas of the genitourinary system due to its role in cell migration and proliferation. It promotes proliferation and migration of muscle cells both in a rat postpartum stress incontinence recovery model and in the rat corpora cavernosa, as well as stimulating differentiation of satellite cells into muscle cells and increasing peri-urethral vascularity [17,42]. These varied impacts of IGF-1 may mean that it is able to support multiple phases of urethral healing in addition to the early healing our protocol is designed to mimic—especially the proliferative phase between days 6 and 10. Overall, IGF-1 appears promising, but further evaluation in vivo will be required to determine the long-term impact of IGF-1 on a urethra healing from injury or surgery.

It is not surprising that VEGF show no significant impact on urothelial wound assays in vitro. The mechanism of VEGF in urethral healing involves promotion of angiogenesis in vivo, as shown by Wang et al. in a rabbit model [38]. Therefore, despite the lack of impact on early urothelial cell growth in our in vitro model, VEGF may still be an important GF for future in vivo study.

TGF-β1 has been highly documented to trigger an EMT [43,44,45]. In the transition, there is an intermediate phase where cells begin by losing their cell-to-cell adhesion and condensing to a smaller, more rounded shape before taking on the more typical mesenchymal morphology. This may explain the significant morphological changes we observed in urothelial cells exposed to TGF-β1. Many of the upregulated genes expressed in our PCR panel are involved in formation of collagen; cell proliferation, differentiation, and motility; and cell shape. Combining the morphology and PCR results, it suggests that EMT is anticipated. At 48 h, the urothelial cells are in the midst of the EMT when the cells lose their adhesion and appear smaller and more rounded, all prior to taking on a mesenchymal phenotype. Despite the potential benefit of TGF-β1 (Table 1), our results suggest it is a poor candidate to support early urethral healing.

There are limitations to this study. The first was the need to use urothelial cells from a bladder source in the wound assays. Many attempts were made by the authors to find a reliable commercial, animal, or human source for urothelial cells from a urethral source. Unfortunately, these sources could not be routinely cultured without fibroblastic transformation. Outside this work, bladder urothelium has been the gold standard of reconstructive techniques for urethral healing or replacement; therefore, this paper is in line with previous studies [27,28,29,30,31,32,33].

The GF concentrations used in this study may not be relevant to those found in human urethras. The goal was to provide support for further tissue engineering work, and the concentrations used were able to show a range of responses and allow for comparative conclusions. The methods of this study to evaluate a single GF at a time are also not comparable to what is found in physiologic urethral healing but were designed to allow for comparison. Combinations of GFs, muscular layers, and 3D systems will all be important directions for future study.

Finally, the protocol described in this study was designed to mimic only the early phase of urethral healing, which occurs via single-layer ingrowth of urothelial cells over a 48 h period. This choice was made given the relative simplicity of healing at this phase and the opportunity for a good in vitro model for comparative evaluation of GFs. While this first step is certainly valuable, further investigation will be needed to understand how GFs impact later healing both in vitro and in vivo. Most importantly, long-term clinical outcomes will be needed to truly understand the impact of any one GF on urethral healing.

## 5. Conclusions

An in vitro protocol evaluating urothelial cell growth using a wound-healing assay designed to mimic early urethral healing successfully allowed for evaluation of the impact of six growth factors. IGF-1 and EGF were both associated with faster cell growth, with IGF-1 showing the most significant difference and EGF showing morphologic changes. Future studies should evaluate how these individual GFs effect urothelial healing in an in vivo model and how they impact later phases of urethral healing.

## Figures and Tables

**Figure 1 mps-06-00064-f001:**
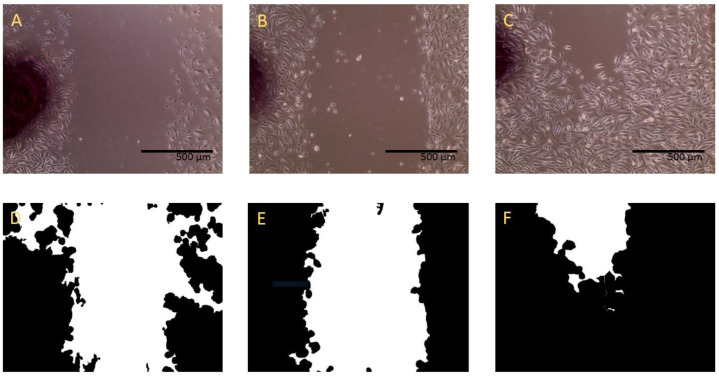
Wound-healing assay and ImageJ calculation. Example of microscopic 10× image results of GF-containing medium scratch assays at time 0 h (**A**), 4 h (**B**), and 24 h (**C**). ImageJ scratch assay analyzer (MiToBo plugin) for time points 0 h (**D**), 4 h (**E**), and 24 h (**F**). Cell growth decreased the total area from 53.8% to 49.0% to 13.0% over the time period.)

**Figure 2 mps-06-00064-f002:**
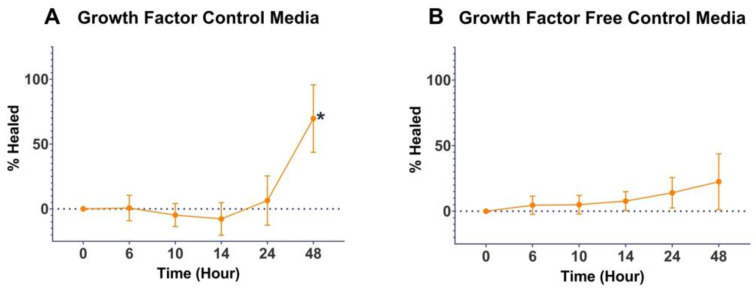
Wound-healing control assay results. Percent healing over 48 h time points of GF-containing medium and GF-free medium. * indicates statically significant percent growth compared to GF-free medium.

**Figure 3 mps-06-00064-f003:**
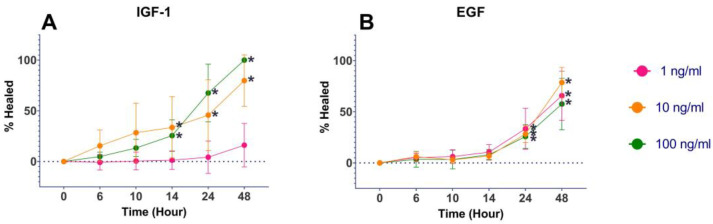
Wound-healing assay results for IGF-1 and EGF. Percent growth over 48 h time points. * indicates statistically significant (<0.05) percent healing compared to GF-free medium control.

**Figure 4 mps-06-00064-f004:**
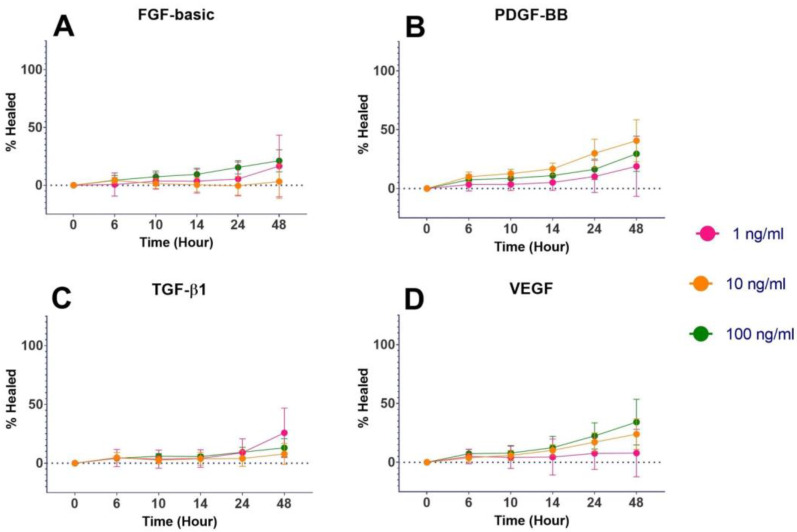
Wound-healing assay results for FGF-basic, PDGF-BB, TGF-β1, and VEGF. Percent healing over 48 h.

**Figure 5 mps-06-00064-f005:**
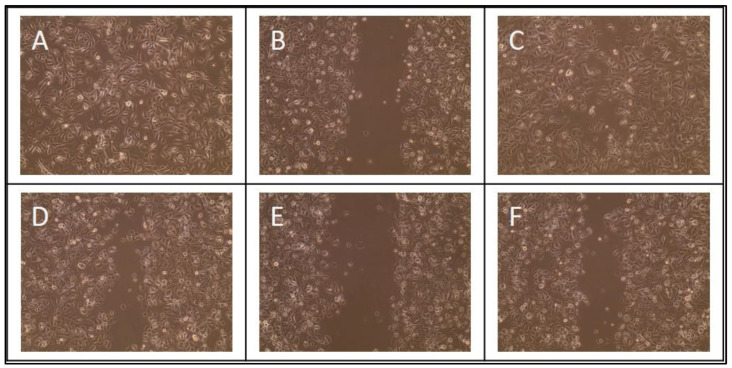
Wound-healing phenotypic changes. Microscopic image of assays containing 10 ng/mL of EGF (**A**), FGF-basic (**B**), IGF-1 (**C**), PDGF-BB (**D**), TGF-β1 (**E**), or VEGF (**F**) at 48 h.

**Figure 6 mps-06-00064-f006:**
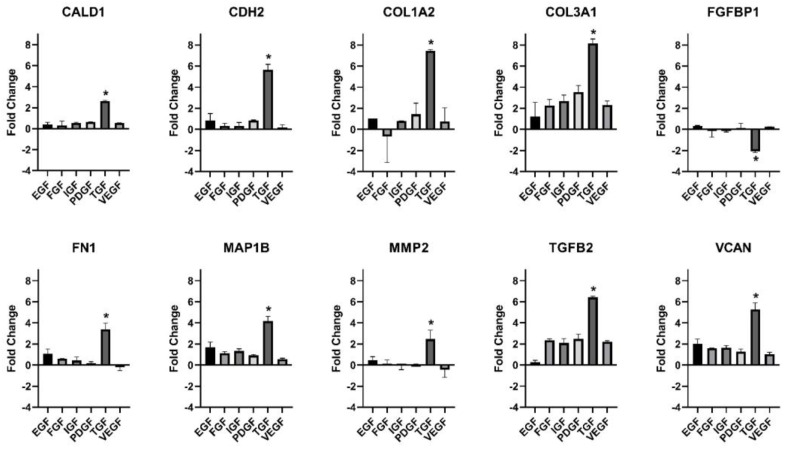
Gene expression representative results. Fold changes relative to GF-containing medium control. * indicates statistically significant (<0.05) differences in fold changes.

**Table 2 mps-06-00064-t002:** Summary of phenotypic changes. Area mean, standard deviation, and form factor for individual GFs compared to GF-containing medium. ***** indicates statistically significant (<0.05) change in form factor compared to GF-containing medium control.

	Area Mean (pixels^2^)	Area Standard Deviation	Form Factor
GF-containing medium control	4737	±1624	0.734
GF-free medium	3307	±1294	0.672
EGF	2722	±1049	0.612 *
FGF-basic	2826	±1584	0.724
IGF-1	3392	±1928	0.729
PDGF-BB	2730	±1357	0.799
TGF-β1	3210	±1813	0.835 *

## Data Availability

The data presented in this study are available on request from the corresponding author.

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
