# Peer review of "Development of a Wound-Healing Protocol for In Vitro Evaluation of Urothelial Cell Growth"

_mps, 2023, doi:10.3390/mps6040064_

Round 1

Reviewer 1 Report

The authors describe research that sought to come up with a new protocol to study urothelial wound healing using various growth factors. This is an important piece of work. However, I came across several errors, both language and formative. I recommend that the authors follow the comments given thus far and make amendments to improve readers‘ interception of their ideas and findings. 

Of note, please weed out phrase urothelial cell healing wherever it appears, and replace with a correct phrase. The study did not assess the healing of cells but of an in vitro model of urothelial healing. 

I recommend that the paper be proofread by a first speaker of English, otherwise be amended to internationally accepted standards. The authors will be invited to resubmit for a final look. 

Development of a Protocol for In Vitro Evaluation of Early Urothelial Cell Healing 

Christopher Foster1, , Todd Jensen1, Christine Finck1,2 and Courtney K. Rowe 1,3,* 4

1 Department of Pediatrics, University of Connecticut School of Medicine, Farmington, CT, USA. 5

2 Division of Pediatric General and Thoracic Surgery, Connecticut Children's, Hartford, CT, USA. 6

3 Division of Pediatric Urology, Connecticut Children's, Hartford, CT, USA. 7

* Correspondence: Crowe@connecticutchildrens.org 8

Abstract: Urethral healing is plagued by strictures and fistulas, impacting quality of life and medical costs. Varieties of growth factors (GFs) have shown promise as a therapeutic approach to improve healing, but to date there has been no protocol for in vitro comparison between GFs. This study focuses on early in vitro urethral healing in response to GFs using a biomimetic urothelial healing assay designed to mimic early in vivo healing. 

Methods: Wound healing assays were developed with human urothelial cells and used to compared six GFs (EGF, FGF-2, IGF-1, PDGF, TGF-β1, VEGF) at three concentrations (1 ng/mL, 10 ng/mL, or 100 ng/mL). Results were compared to commercially available GF-containing media and GF-free media over a 48-hour period. 

Results: IGF-1 at concentrations of 10 and 100 ng/mL had a statistically significant increase (p<0.0002) in healing compared to controls. EGF at all concentrations of 1, 10, and 100 ng/mL had a statistically significant increase (p<0.0002) in healing compared to the GF-free media control. Conclusion: This study is the first to use a clinically relevant protocol for wound healing to evaluate urothelial healing, and the first to compare GFs for future use in reconstructive techniques to improve urethral healing. 

Comment on the Title

Development of a Protocol for In Vitro Evaluation of Early Urothelial Cell Healing. Does your protocol show cell healing or in vitro wound healing in a urothelial cell model? Rephrase.

Comments on the Abstract

1. Urethral healing is plagued by strictures and fistulas, impacting quality of life and medical costs. Meaning? Amend this statement.

2. Wound healing assays were developed with human urothelial cells and used to compared six GFs (EGF, FGF-2, IGF-1, PDGF, TGF-β1, VEGF) at three concentrations (1 ng/mL, 10 ng/mL, or 100 ng/mL). Results were compared to commercially available GF-containing media and GF-free media over a 48-hour period. What is the relationship between the six IGFs used and those in media? Not clear here.

3. Wound healing assays were developed with human urothelial cells and used to compared six GFs (EGF, FGF-2, IGF-1, PDGF, TGF-β1, VEGF) at three concentrations. .... to compared, amend. 

4. IGF-1 at concentrations of 10 and 100 ng/mL had a statistically significant increase (p<0.0002) in healing compared to controls.. .... had a statistically significant increase, amend.

5. This study is the first to use a clinically relevant protocol for wound healing to evaluate urothelial healing, and the first to compare GFs for future use in reconstructive techniques to improve urethral healing. The results of this study are only valid for a urothelial cell culture. Such results may not translate to ‘urothelial wound healing’ as implied here. Be careful with such insinuations. 

Comments on Introduction 

...revise sompound statement....and leading to significant medical costs

Regenerative medicine approaches have used growth factors (GFs) to provide enticing preliminary options for urethral tissue replacement and repair. GFs such as VEGF, IGF-1, EGF, mitomycin C, and TGF-B appear to improve urethral healing in animal and in vitro models, but the study methods have been so varied that there is no consensus on the type of GF, concentration, location of delivery or timing of delivery needed to impact urethral healing.

... Regenerative medicine approaches...revise

GFs such as VEGF, IGF-1, EGF, mitomycin C, and TGF-B appear to improve urethral healing in animal and in vitro models...provide references 

but the study methods have been so varied that there is no consensus on the type of GF, concentration, location of delivery or timing of delivery needed to impact urethral healing...revise

The goal of this study was to develop a protocol for evaluating the differences in early in vitro urethral healing rates between six GFs known to be involved in urethral development or healing (Table 1) using a wound healing assay. Wound healing assays are a useful, simple and well-established technique for studying the migration of cells in vitro.

evaluating the differences in early in vitro urethral healing rates between six GFs known to be involved in urethral development or healing...revise

Wound healing assays are a useful, simple and well-established technique for studying the migration of cells in vitro. This may require a touch up.

A wound or scratch is created in a cellular layer after which cells from the edges will migrate and fill in the gap. At the initial and later time points, images are compared to determine the rate of healing. Healing is defined as the percent of wound gap covered by urothelial cells. This technique was chosen as the best in vitro strategy to mimic in vivo urethral healing, which was investigated by Lopes et al in pigs, Hafez et al in rabbits, and Hofer in rats.

A wound or scratch is created in a cellular layer after which cells from the edges will migrate and fill in the gap... revise. Expand and be concise. Also read the whole paragraph at once and make the ideas flow. In its current for, the paragraph remains incoherent and incomplete.

These studies found that the urethra first heals after an injury or surgery via ingrowth from the edges of the urothelial defect in a single layer over the first 48 hours.3-5 This pattern is mimicked by the protocol described below, allowing for a clinically relevant experimental model to evaluate the impact of various GFs on early urethral cell healing and serve as a backbone of future regenerative medicine techniques. 

These studies...which studies? Be sure to include references where necessary throughout your manuscript.

This pattern is mimicked by the...which pattern are you referring to here? Incoherent statements.

evaluate the impact of various GFs on early urethral cell healing... revise impact and cell healing. Your study was not assessing cell healing.

2. Materials and Methods

2.1. Cell Expansion...revise

Primary Bladder Epithelial Cells (ATCC, Manassas, VA) were purchased and expanded using the provided protocol. Bladder cells are a very common and available substitute used in urethral research. Briefly, cells were transferred to a T75 flask at 4000 cells per cm2 with Prostate Epithelial Basal Medium (ATCC, Manassas, VA) supplemented with Corneal Epithelial Cell Growth Kit (ATCC, Manassas, VA) and Penicillin-Streptomycin-Amphotericin B Solution (ATCC, Manassas, VA) at 5 mL of complete growth media per 25 cm2

Bladder cells are a very common and available substitute used in urethral research. Avoid use of very... revise. What does very common mean?

. Every 48 hours, spent media was removed and fresh media was added until approximately 80% confluence was reached. Upon reaching approximately 80% confluence, spent media was removed and cell layer was detached using 0.05% Trypsin-EDTA. Trypsin-EDTA was neutralized with equal volume FBS. Cells were collected and centrifuged for 5 minutes at 150g ...need to specify the equipment here.. Cells were counted and either frozen in working volumes with 10% DMSO or used for wound healing assay at passage 4. What does this citation imply here?

2.2. Selection of Evaluated GFs and Controls 

Six candidate GFs were selected based on their role in urethral or wound healing and their presence in the rat urethra during healing after surgery (Table 1).5 Revise 5, is it from one reference? Each GF was added directly to cell media as described below. GF concentrations were chosen based on preliminary trials within our lab to allow demonstration of a wide range of dose-dependent responses.

Two controls were used. GF-containing control media describes the commercially available media and growth kit that is instructed to use by American Type Culture Collection for growth of their cells. Specify the media and its source. This growth kit contains EGF, TGF-α, KGF, and Extract P; specific quantities are proprietary. We also use a GF-free control media to have a baseline that does not contain any GFs. Which kit?

2.3. Wound Healing Assay

Human urothelial cells from a bladder source, at a concentration of 6x105 per mL of media (42,000 cells for 70 µL total of media) were inserted into all 3 wells of a wound healing assay culture- insert (ibidi Inc, Fitchburg, WI). Cells were allowed to reach near confluence for approximately 20 hours. 4 hours prior to start of assay, cells were rinsed and 70 µL of fresh CnT-Prime Basal, Growth

Human urothelial cells from a bladder source...bladder source meaning?

into all 3 wells of a wound healing assay culture- insert...all three wells or just three wells? Revise.

Cells were allowed to reach near confluence for approximately 20 hours. Think about reach near. Revise.

4 hours prior to start of assay, cells were rinsed and 70 µL of fresh CnT-Prime Basal, Growth Factor Free Media (Zenbio Inc, Research Triangle Park, NC) was added into the wells....Revise. 

At time 0 of assay, spent media was aspirated and the well insert was removed from the dish. The dish was rinsed with PBS and 2 mL of fresh media (Zenbio growth factor free or control ATCC epithelial basal media) was added to dish. GFs EGF, FGF-basic, IGF-1, PDGF-BB, TGF-β1, and VEGF (Pepro- Tech, Rocky Hill, NJ) were added to their respective tests at concentrations of 1 ng/mL, 10 ng/mL, or 100 ng/mL. Concentration ranges were determined through previous trials and chosen as the best representation....Revise

3. Results

GF-containing control media has a statistically significant increase (p<0.0002)

Revise the rest of the sections the same way.

The manuscript requires extensive review as some of the ideas are confounded by inappropriate use of phrases and terms. I suggest the manuscript be taken for professional English review. I can run through the manuscript and recommend publications once I will be satisfied that appropriate measures have been taken to make the presentation as clear and correct as possible.

Reviewer 2 Report

Overview of the manuscript
The article focuses on presenting the methods and results of the explorative study, in which bladder epithelial cells were treated with different growth factors. Methods such as wound healing assay, PCR analysis, and morphological change analysis of cells were adopted in the present study. The authors found that, among the several growth factors tested, IGF-1 and EGF were shown to significantly increase cellular wound healing, whereas TGF-
β1 showed several significant changes in gene expression. The authors propose the methodological protocol used in the present work as clinically relevant for future use in reconstructive techniques to improve urethral healing.

 GENERAL COMMENT

The work does not present significant novelty results. The protocol adopted are not new and the experimental plan remains inconsistent related to the aim expressed in the title. The authors perform their work addressing the reader to the evaluation of urethral healing, but they use bladder epithelial cells. The authors do not attempt to follow protocols for the use of urethral cells as other authors, cited in reference, do. Morphological analysis is well performed but of limited value in providing an adequate assessment of cell change. Furthermore, a comparison with different cell lines, from other tissues, should have been more interesting to highlight the specific effect of tested GF on urethral cells or, as in this case, bladder cell taken as model in spite of urethral cells. The statistic applied is not adequate or not adequately explained.

SPECIFIC COMMENTS

Title

The title is not properly focused on your work, you use bladder epithelial cell line. Change it.

Abstract

The abstract should be revised, no indication about PCR results and morphological cell analysis are reported.

Materials and Methods

Pag 2, line 51: the use of only Primary Bladder Epithelial Cells, in your work, where the goal is the urethral wound healing, is not adequate or sufficient to obtain valid results.

Pag. 3, line 91-94: Statistical details are insufficient.  2-way ANOVA is used when the source of variance is double. What are your sources of variance? Explicit here the multiple comparison tests used, not only in the figure legend, and explain why you have used them.

Results

Pag. 6, line 161-165: explain what type of genes you have analysed and why have you chosen them.

Discussion

Pag. 7, line 177: you did not investigate urothelial cell but bladder epithelial cells. Change the sentence.

Pag. 7, line 199-204: The paragraph is useless. For what reason do you explain the embryogenesis of urethra? Delete it or give your reasons.

 Pag. 8, line 274-278: the conclusions are devoid of significance. You did not use a real wound healing urethral model to propose. The administration of growth factors is not accompanied by an adequate and sufficient analysis of the results convincingly addressing the use of one or more GF for urethral wound healing.

none

Round 2

Reviewer 1 Report

Review Report

Title: Development of a Wound Healing Protocol for In Vitro Evaluation of Urothelial Cell Growth

Abstract:

Revise this statement: Cell growth was 15 compared to commercial GF-containing media (EGF, TGF-α, KGF, Extract P) and GF-free media 16 over a 48-hour period. Check the same issue in the Results and Discussion sections.

Revise these statements: There was a statistically significant increase in cell growth for IGF-1 17 at 10 and 100 ng/mL compared to both controls. There was a statistically significant increase in cell 18 growth for EGF at all concentrations compared to the GF-free media control. Include a measure of significance here.

The manuscript is now in acceptable state. I recommend that the authors read through and submit for final submission.

Review Report

Title: Development of a Wound Healing Protocol for In Vitro Evaluation of Urothelial Cell Growth

Abstract:

Revise this statement: Cell growth was 15 compared to commercial GF-containing media (EGF, TGF-α, KGF, Extract P) and GF-free media 16 over a 48-hour period. Check the same issue in the Results and Discussion sections.

Revise these statements: There was a statistically significant increase in cell growth for IGF-1 17 at 10 and 100 ng/mL compared to both controls. There was a statistically significant increase in cell 18 growth for EGF at all concentrations compared to the GF-free media control. Include a measure of significance here.

Reviewer 2 Report

The previous concerns have been solved.

No more concerns.

Author Response

Thank you for your input.